# Shifted Chunk Transformer for
# Spatio-Temporal Representational Learning

**Xuefan Zha**
Kuaishou Technology
zhaxuefan@kuaishou.com

**Wentao Zhu**
Kuaishou Technology
wentaozhu@kuaishou.com

**Tingxun Lv**
Kuaishou Technology
lvtingxun@kuaishou.com

**Sen Yang**
Kuaishou Technology
senyang@kuaishou.com

**Ji Liu**
Kuaishou Technology
ji.liu.uwisc@Gmail.com

## Abstract

Spatio-temporal representational learning has been widely adopted in various fields such as action recognition, video object segmentation, and action anticipation. Previous spatio-temporal representational learning approaches primarily employ ConvNets or sequential models, *e.g.*, LSTM, to learn the *intra-frame* and *inter-frame* features. Recently, Transformer models have successfully dominated the study of natural language processing (NLP), image classification, etc. However, the pure-Transformer based spatio-temporal learning can be prohibitively costly on memory and computation to extract *fine-grained* features from a tiny patch. To tackle the training difficulty and enhance the spatio-temporal learning, we construct a *shifted chunk Transformer* with pure self-attention blocks. Leveraging the recent efficient Transformer design in NLP, this shifted chunk Transformer can learn hierarchical spatio-temporal features from a local tiny patch to a global video clip. Our shifted self-attention can also effectively model complicated *inter-frame* variances. Furthermore, we build a clip encoder based on Transformer to model long-term temporal dependencies. We conduct thorough ablation studies to validate each component and hyper-parameters in our shifted chunk Transformer, and it outperforms previous state-of-the-art approaches on Kinetics-400, Kinetics-600, UCF101, and HMDB51.

## 1 Introduction

Spatio-temporal representational learning tries to model complicated *intra-frame* and *inter-frame* relationships, and it is critical to various tasks such as action recognition [20], action detection [55, 52], object tracking [23], and action anticipation [21]. Deep learning based spatio-temporal representation learning approaches have been largely explored since the success of AlexNet on image classification [24, 11]. Previous deep spatio-temporal learning can be mainly divided into two aspects: deep ConvNets based methods [35, 15, 16] and deep sequential learning based methods [55, 28, 29]. Deep ConvNets based methods are primarily integrated various factorization techniques [51, 34], or a *priori* [16] for efficient spatio-temporal learning [15]. Some works focus on extracting effective spatio-temporal features [41, 8] or capturing complicated long-range dependencies [49]. Deep sequential learning based methods try to formulate the spatial and temporal relationships through advanced deep sequential models [28] or the attention mechanism [29].

On the other hand, the Transformer has become the *de-facto* standard for sequential learning tasks such as speech and language processing [44, 12, 54, 19]. The great success of Transformer on natural language processing (NLP) has inspired computer vision community to explore self-attention

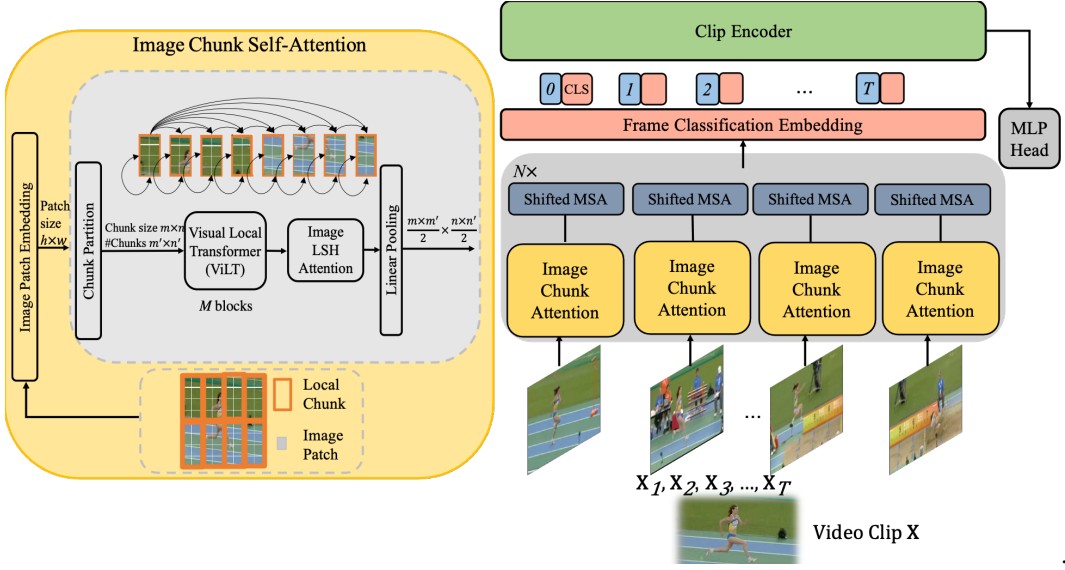

Figure 1: The framework of the proposed shifted chunk Transformer which involves two main components, frame encoder (dark grey) and clip encoder. The frame encoder consists of $N$ alternative blocks of image chunk self-attention (left) and shifted multi-head self-attention (MSA).

structures for several vision tasks, *e.g.*, image classification [13, 40, 36], object detection [6], and super-resolution [33]. The main difficulty in pure-Transformer models for vision is that Transformers lack the inductive biases of convolutions, such as translation equivariance, and they require more data [13] or stronger regularisation [40] in the training. It is only very recently that, vision Transform (ViT), a pure Transformer architecture, has outperformed its convolutional counterparts in image classification when pre-trained on large amounts of data [13]. However, the hurdle is aggravated when the pure-Transformer design is applied to spatio-temporal representational learning.

Recently, a few attempts have been made to design pure-Transformer structures for spatio-temporal representation learning [4, 5, 14, 2]. Simply applying Transformer to 3D video domain is computationally intensive [4]. The Transformer based spatio-temporal learning methods primarily focus on designing efficient variants by factorization along spatial- and temporal-dimensions [4, 5], or employing a multi-scale pyramid structure for a trade-off between the resolution and channel capacity while reducing the memory and computational cost [14]. The spatio-temporal learning capacity can be further improved by extracting more effective fine-grained features through advanced and efficient *intra-frame* and *inter-frame* representational learning.

In this work, we propose a novel spatio-temporal learning framework based on pure-Transformer, called shifted chunk Transformer as illustrated in Fig. 1, which extracts effective fine-grained *intra-frame* features with a low computational complexity leveraging the recent advance of Transformer in NLP [22]. Specially, we divide each frame into several local windows called image chunks, and construct a hierarchical image chunk Transformer, which employs locality-sensitive hashing (LSH) to enhance the dot-product attention in each chunk and reduces the memory and computation consumption significantly. To fully consider the motion effect of object, we design a robust self-attention module, shifted self-attention, which explicitly extracts correlations from nearby frames. We further design a pure-Transformer based frame-wise attention module, clip encoder, to model the complicated *inter-frame* relationships with a minimal extra computational cost. Our contributions can be summarized as follows:

- We construct an image chunk self-attention to mine fine-grained *intra-frame* features leveraging the recent advance of Transformer. The hierarchical image chunk Transformer employs locality-sensitive hashing (LSH) [3] to reduce the memory and computation consumption significantly, which enables an effective spatio-temporal learning directly from a tiny patch.

- We build a shifted self-attention to fully consider the motion effect of objects, which yields effective modeling of complicated *inter-frame* variances in the spatio-temporal representa-

tional learning. Furthermore, a clip encoder with a pure-Transformer structure is employed for frame-wise attention, which models complicated and long-term *inter-frame* relationships at a minimal extra cost.

- The shifted chunk Transformer with pure-Transformer outperforms previous state-of-the-art approaches on several action recognition benchmarks, including Kinectics-400 [20], Kinetics-600 [7], UCF101 [39] and HMDB51 [25].

## 2  Related Work

**Conventional deep learning based action recognition**  Conventional deep spatio-temporal representational learning mainly involves two aspects: deep sequential learning based methods [55, 28, 29] and deep ConvNet based methods [35, 15, 16]. The recurrent networks can be extended to 3D spatio-temporal domain for action recognition [28]. In deep ConvNet based methods, two-stream ConvNet employs two branches of 2D ConvNets and explicitly models motion by optical flow [38]. The C3D [41] and I3D [8] directly extend 2D ConvNets to 3D ConvNets, which is natural for 3D spatio-temporal representational learning [9]. However, the 3D ConvNet requires significantly more computation and more training data to achieve a desired accuracy [51]. Thus, P3D [34] and S3D [51] attempt to factorize the 3D convolution into a 2D spatial convolution and a 1D temporal convolution. SlowFast network [16] and X3D [15] conduct trade-offs among resolution, temporal frame rate and the number of channels for the efficient video recognition. Non-local network [49] proposes to add non-local operations in deep network and captures long-range dependencies. The recent pure-Transformer based spatio-temporal learning enables longer dependency relationship modeling and further increases the accuracy of action recognition [4].

**Vision Transformers**  NLP community has witnessed the great success of pre-training by Transformer [44, 12], and it has been emerging for image classification [13, 40, 36], object detection [6], and image super-resolution [33]. CPVT [10] employs pre-defined and independent input tokens to increase the generalization for image classification. Pure-Transformer network has no inductive bias or prior as ConvNets. ViT [13] pre-trains on large amounts of data and attains excellent results on image classification. CvT [50] introduces convolutions into ViT to yield better performance and efficiency. DeiT [40] and MViT [14] instead employ distillation and multi-scale to cope with the training difficulty. PVT [47] and segmentation Transformer [53] further extend Transformer to dense prediction tasks, *e.g.*, object detection and semantic segmentation. Simply applying Transformer to 3D video spatio-temporal representational learning aggravates the training difficulty significantly, and it requires advanced model design for pure-Transformer based spatio-temporal learning.

**Transformer based action recognition**  Recently, only a few works have been conducted using pure-Transformer for spatio-temporal learning [4, 14, 5, 2]. Most of the efforts focus on designing efficient Transformer models to reduce the computation and memory consumption. ViViT [4] and TimeSformer [5] study various factorization methods along spatial- and temporal-dimensions. MViT [14] conducts a trade-off between resolution and the number of channels, and constructs a multi-scale Transformer to learn a hierarchy from simple dense resolution and fine-grained features to complex coarse features. VATT [2] conducts unsupervised multi-modality self-supervised learning with a pure-Transformer structure. In this work, we extract fine-grained *intra-frame* features from each tiny patch and model complicated *inter-frame* relationship through efficient and advanced self-attention blocks.

## 3  Method

In this section, we describe each component of our shifted chunk Transformer for spatio-temporal representation learning in video based action recognition.

### 3.1  Overview

Let $\mathbf{X} = [\mathbf{X}_1, \mathbf{X}_2, \cdots, \mathbf{X}_T] \in \mathbb{R}^{T \times H \times W \times 3}$ be one input clip of $T$ RGB frames sampled from a video, where $\mathbf{X}_i \in \mathbb{R}^{H \times W \times 3}$ is the $i$-th frame in the clip, and $H$ and $W$ are the frame size. To design an efficient pure-Transformer based spatio-temporal learning, we construct a shifted chunk Transformer, including image chunk self-attention blocks, shifted multi-head self-attention blocks,

and a clip encoder, as illustrated in Fig. 1. We first construct an image chunk self-attention block leveraging the advanced efficient Transformer design in NLP [22], which is illustrated in the left of Fig. 1. The locality-sensitive hashing (LSH) [3] in the image chunk self-attention enables a relatively small patch as a token, thus it is capable of extracting fine-grained *intra-frame* features. A linear pooling layer is designed to adaptively reduce the resolution after LSH attention. After that, a shifted self-attention is designed to extract motion related *inter-frame* features. Our shifted self-attention considers the motion of objects across nearby frames and explicitly models the temporal relationship into self-attention. The frame encoder shown in dark grey color can be an effective feature extractor which can be stacked for several times. The hierarchical frame encoder and image chunk self-attention further learns an effective multi-level feature from local to global abstraction. Lastly, we employ a pure-Transformer to learn complicated *inter-frame* relationships and frame-wise attention along the temporal dimension. We use multi-head self-attention (MSA) in all the blocks.

## 3.2 Image Chunk Self-Attention

Transformer can learn complicated long range dependencies which can be computed through the high efficient matrix product [44]. Different from convolution which has inherent inductive bias [13], Transformer learns the entire features from data. The main challenge for a pure-Transformer based vision model mainly involves two aspects: 1) how to design an efficient model to learn effective features from the entire image,

Table 1: ViT-B [13] with a smaller patch size yields higher accuracy (%).

| Crop size | Patch size | K400 | UCF101 |
|---|---|---|---|
| 224 | 16 | 75.3 | 95.3 |
| 224 | 8 | **78.4** | **97.0** |

because simply treating each pixel as a token is computationally intensive, 2) how to train this powerful model and learn various effective features from data. ViT [13] treats each patch of size $16 \times 16$ as one token and pre-trains the model with large amounts of data. We argue that Transformer with a smaller patch as a token can extract fine-grained features which improves spatio-temporal learning for action recognition. For ViT-B-16 [13] of crop size $224 \times 224$ as a frame encoder followed by a shifted MSA and a clip encoder, a smaller patch size of $8 \times 8$ yields better accuracy on Kinetics-400 and UCF101 as shown in Table 1.

The Transformer computes each pairwise correlation through a dot product, thus it has a high computational complexity of $O(L^2)$, where $L$ is the totally number of tokens. In natural language processing (NLP), LSH attention [22] employs locality-sensitive hashing (LSH) bucketing [3] and chunk sorting for queries and keys to approximate the attention matrix computation. Leveraging the efficient LSH approximation, the LSH attention reduce the computation complexity to $O(L \log L)$.

To preserve locality property and learn transition and rotation invariant low-level features from images, we firstly design a visual local transformer (ViLT) with shared parameters along different image local windows, or image chunks. Each image chunk consists of multiple tiny patches as illustrated in the left bottom block of Fig. 1. We intend to employ patches of a small size in the ViLT which is the first level abstraction of the input, so that the model extracts a fine-grained representation which enhances the entire spatial-temporal learning. Employing the small patch yields large number of tokens in the following self-attention, and we construct an image locality-sensitive hashing (LSH) attention leveraging advanced design of Transformer in NLP [22]. The image LSH attention can efficiently extract higher-level and plentiful *intra-frame* features ranging from a tiny patch to the entire image. The framework of image chunk self-attention is illustrated in the left part of Fig. 1.

**Visual local transformer (ViLT)** In the shifted chunk Transformer, we firstly construct a ViLT which slides one self-attention for each tiny patch along the whole image. The ViLT is illustrated in the bottom block of Fig. 1. Let $h$, $w$ be the height and width of the tiny image patch $\mathbf{p} \in \mathbb{R}^{h \times w \times 3}$. Following the success of ViT [13], we treat each patch as one dimensional vector of length $h \times w \times 3$. Suppose each chunk consists of $m \times n$ tiny patches, denoted as $\{\mathbf{p}_{1,1}, \mathbf{p}_{1,2}, \cdots, \mathbf{p}_{1,n}; \mathbf{p}_{2,1}, \cdots, \mathbf{p}_{2,n}; \cdots; \mathbf{p}_{m,1}, \cdots, \mathbf{p}_{m,n}\}$. After flattening the chunk into a list of tiny patches, we denote the chunk as $\{\mathbf{p}_1, \mathbf{p}_2, \cdots, \mathbf{p}_L\}$ without loss of generality, where $L = m \times n$.

In ViLT, we use a learnable 1D position embedding $\mathbf{E}_{pos} \in \mathbb{R}^{L \times D}$ to retain position information

$$\mathbf{z}_0 = [\mathbf{p}_1 \mathbf{E}; \mathbf{p}_2 \mathbf{E}; \cdots; \mathbf{p}_{m \times n} \mathbf{E}] + \mathbf{E}_{pos}, \quad \mathbf{E} \in \mathbb{R}^{(h \times w \times 3) \times D}, \tag{1}$$

where $\mathbf{E}$ is the linear patch embedding matrix, and $D$ is the embedding dimension. Then we can construct alternating layers of multi-head self-attention (MSA), MLP with GELU [18] non-linearity, Layernorm (LN) and residual connections [44] for the chunk as

$$\mathbf{z}'_m = \text{MSA}(\text{LN}(\mathbf{z}_{m-1})) + \mathbf{z}_{m-1}, \quad \mathbf{z}_m = \text{MLP}(\text{LN}(\mathbf{z}'_m)) + \mathbf{z}'_m, \quad m = 1, \cdots, M, \quad (2)$$

where $M$ is the number of blocks. We conduct the ViLT sliding the entire image without over-lapping. The parameters of the ViLT are shared among all the image chunks, which forces the chunk self-attention to learn translation and rotation invariant, and fine-grained features. The tiny patch-wise feature extraction preserves the locality property, which is a strong prior for natural images. After the ViLT, we obtain the extracted features for the entire image denoted as $\mathbf{y} = [\mathbf{y}_1; \mathbf{y}_2; \cdots; \mathbf{y}_L; \cdots; \mathbf{y}_{L \times L'}]$, where the entire image can be split into $L' = m' \times n' = \lceil \frac{H}{h \times m} \rceil \times \lceil \frac{W}{w \times n} \rceil$ chunks, and we conduct zero padding for the last chunks in each row and column.

The ViLT forces to learn image locality features which is a desired property for a low-level feature extractor [26]. For pure-Transformer based vision system, it also reduces the memory consumption significantly because it restricts the correlation of one tiny patch within the local chunk. Therefore, the memory and computational complexity of dot-product attention in ViLT can be reduced to $O(L^2)$ compared to the complexity of conventional self-attention $O((L' \times L)^2)$.

**Image locality-sensitive hashing (LSH) attention**  After ViLT blocks, we obtain local fine-grained features of length $L' \times L$. Since the patch size is tiny, the total number of patches can be large, which leads to more difficult training than other vision Transformers [4, 13]. On the other hand, the problem of finding nearest neighbors quickly in high-dimensional spaces can be solved by locality-sensitive hashing (LSH), which hashes similar input items into the same "buckets" with high probability. In NLP, LSH attention [22] is proposed to handle quite long sequence data, which employs locality-sensitive hashing (LSH) bucketing approximation and bucket sorting to reduce the computational complexity of matrix product between query and key in self-attention.

In dot-product attention, the softmax activation function pushes the attention weights close to 1 or 0, which means the attention matrix is typically sparse. The query and key can be approximated by locality-sensitive hashing (LSH) [3] to reduce the computational complexity. Furthermore, through bucketing sort, the attention matrix product can be accelerated by a chunk triangular matrix product, which has been validated by LSH attention [22].

The used multi-head image LSH attention can be constructed as

$$\mathbf{y}' = \text{MSA}(\text{LSHAtt}(\text{LN}(\mathbf{y}))) + \mathbf{y}, \quad \mathbf{s} = \text{MLP}(\text{LN}(\mathbf{y}')) + \mathbf{y}', \quad (3)$$

where LSH attention $\text{LSHAtt}(\cdot)$ employs angular distance to conduct LSH hashing [3]. The image LSH attention reduces the memory and time complexity to $O(L' \times L \log(L' \times L))$, compared with that of conventional dot-product attention $O((L' \times L)^2)$. The image LSH attention reduces the complexity significantly because the patch size is tiny and the number of tiny patches $L' \times L$ are large. The image level LSH attention in the second level learns relatively global features from the first level's local fine-grained features.

The hierarchical feature learning from local to global has been validated as an effective principle for vision system design [26, 45]. Inspired by hierarchical abstraction in ConvNets [26], we construct a linear pooling layer which firstly conducts squeeze then employs linear projection for feature dimension reduction. The linear pooling adaptively squeezes the sequence length by $\frac{1}{4}$.

$$\text{Reshape:} \quad \mathbf{s} = [\mathbf{s}_{1,1}, \cdots, \mathbf{s}_{1,n \times n'}; \cdots; \mathbf{s}_{m \times m',1}, \cdots, \mathbf{s}_{m \times m',n \times n'}],$$

$$\text{Squeeze:} \quad \mathbf{s}' = [\mathbf{s}'_{1,1}, \cdots, \mathbf{s}'_{1,n \times n'/2}; \cdots; \mathbf{s}'_{m \times m'/2,1}, \cdots, \mathbf{s}'_{m \times m'/2,n \times n'/2}], \quad (4)$$

$$\text{Linear:} \quad \mathbf{t} = [\mathbf{s}'_{1,1}\mathbf{E}_t, \cdots, \mathbf{s}'_{1,n \times n'/2}\mathbf{E}_t; \cdots; \mathbf{s}'_{m \times m'/2,1}\mathbf{E}_t, \cdots, \mathbf{s}'_{m \times m'/2,n \times n'/2}\mathbf{E}_t],$$

where $\mathbf{s}'_{i,j}$ in $\mathbf{s}'$ concatenates $\mathbf{s}_{2i-1,2j-1}$, $\mathbf{s}_{2i-1,2j}$, $\mathbf{s}_{2i,2j-1}$ and $\mathbf{s}_{2i,2j}$ from $\mathbf{s}$, $\mathbf{E}_t$ is the linear projection matrix to adaptively reduce the number of dimensions after squeeze by half. The reshape is to retain the spatial relationship of each patch, and the squeeze reduces the number of patches by $\frac{1}{4}$ and enlarges the feature dimensions by four times. The linear pooling layer forces the model to learn high-level global features in the following layers.

## 3.3  Shifted Multi-Head Self-Attention

Considering the motion effect of objects, we explicitly construct a shifted multi-head self-attention (MSA) for spatio-temporal learning after image chunk self-attention as illustrated in Fig. 1 and Fig. 2.

For video classification, a special classification token ([CLS]) [12] can be prepended into the feature sequence. To learn frame-wise spatio-temporal representations, we prepend the [CLS] token to each frame. Without loss of generality, we denote the image chunk self-attention feature to be a list of $[\mathbf{t}_{i,1}; \mathbf{t}_{i,2}; \cdots ; \mathbf{t}_{i,L \times L'/4}]$ for the $i$-th frame. Then, we obtain the input of $T$ frames for shifted MSA as

$$
\begin{aligned}
\mathbf{t}' &= [\mathbf{t}'_{1,1}; \cdots ; \mathbf{t}'_{1,1+L \times L'/4}; \cdots ; \mathbf{t}'_{T,1}; \cdots ; \mathbf{t}'_{T,1+L \times L'/4}] \\
&= [\mathbf{z}_{1,cls}; \mathbf{t}_{1,1}; \cdots ; \mathbf{t}_{1,L \times L'/4}; \cdots ; \mathbf{z}_{T,cls}; \mathbf{t}_{T,1}; \cdots ; \mathbf{t}_{T,L \times L'/4}].
\end{aligned} \tag{5}
$$

The shifted multi-head self-attention explicitly considers the *inter-frame* motion of objects, which computes the correlation between the current frame and the previous frame in the attention matrix, which can be formulated as

$$
\mathbf{q}^i_{t,p} = \text{LN}(\mathbf{t}'_{t,p})\mathbf{W}^i_Q, \quad \mathbf{k}^i_{t,p} = \text{LN}(\mathbf{t}'_{t-1,p})\mathbf{W}^i_K, \quad \mathbf{v}^i_{t,p} = \text{LN}(\mathbf{t}'_{t,p})\mathbf{W}^i_V, \tag{6}
$$

where $\mathbf{W}^i_Q$, $\mathbf{W}^i_K$ and $\mathbf{W}^i_V$ are projection matrices for head $i$, and we employ a cyclic way to calculate the key of the first frame $t = 1$ by defining $\mathbf{t}'_{0,p} = \mathbf{t}'_{T,p}$. By concatenating $\mathbf{q}^i_{t,p}$, $\mathbf{k}^i_{t,p}$ $\mathbf{v}^i_{t,p}$ into matrices $\mathbf{Q}^i_t$, $\mathbf{K}^i_t$, $\mathbf{V}^i_t$ along patch location $p$, the shifted MSA for frame $t$ can be calculated as

$$
\begin{aligned}
\mathbf{a}_t &= \text{Concat}(\text{Attention}(\mathbf{Q}^1_t, \mathbf{K}^1_t, \mathbf{V}^1_t), \cdots , \text{Attention}(\mathbf{Q}^l_t, \mathbf{K}^l_t, \mathbf{V}^l_t))\mathbf{W}_O, \\
\mathbf{a}'_t &= \text{MLP}(\text{LN}(\mathbf{a}_t)) + \mathbf{a}_t,
\end{aligned} \tag{7}
$$

where $l$ is the number of heads in multi-head self-attention. The shifted self-attention compensates object motion and spatial variances. We explicitly integrate motion shift into self-attention, which extracts robust features for spatio-temporal learning. The block with alternating layers of image chunk self-attention and shifted MSA can be stacked for multiple times to fully extract effective hierarchical fine-grained features from tiny local patches to the whole clip in our shifted chunk Transformer.

### 3.4 Clip Encoder for Global Clip Attention

To learn complicated *inter-frame* relationship from the extracted frame-level features, we design a clip encoder based on a pure-Transformer structure to adaptively learn frame-wise attention. To facilitate the video classification, we prepend a global special classification token ([CLS]) into the frame-level feature sequence.

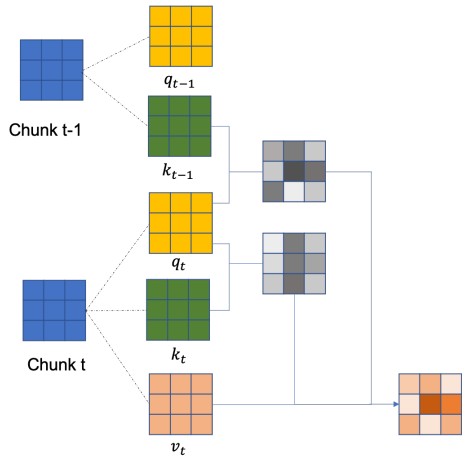

Figure 2: Illustration of shifted MSA, which explicitly extracts fine-grained motion information along two frames.

In this module, we employ the classification feature $\mathbf{a}'_{t1}$ corresponding to $\mathbf{z}_{t,cls}$ as the frame-level feature for frame $t$. To consider frame position, we also employ a standard learnable 1D position embedding as each frame position embedding. The clip encoder can be formulated as

$$
\begin{aligned}
\mathbf{b}_0 &= [\mathbf{b}_{cls}; \mathbf{a}'_{11}\mathbf{E}'; \cdots ; \mathbf{a}'_{T1}\mathbf{E}'] + \mathbf{E}'_{pos}, \quad \mathbf{E}'_{pos} \in \mathbb{R}^{(T+1) \times D'}, \\
\mathbf{b}'_m &= \text{MSA}(\text{LN}(\mathbf{b}_{m-1})) + \mathbf{b}_{m-1}, \quad \mathbf{b}_m = \text{MLP}(\text{LN}(\mathbf{b}'_m)) + \mathbf{b}'_m, \quad m = 1, \cdots , M', \\
\mathbf{c} &= \text{MLP}(\text{LN}(\mathbf{b}_{M'1}))
\end{aligned} \tag{8}
$$

where $\mathbf{E}'$ is the linear frame embedding matrix, $D'$ is the clip encoder embedding size, $M'$ is the number of blocks, and $\mathbf{b}_{M'1}$ is the clip-level classification feature for the classification token $\mathbf{b}_{cls}$, $\mathbf{c}$ is the video classification logit for softmax. We use dropout [24] for the second last layer and cross-entropy loss with label smoothing [32] for training. The clip encoder can be efficient with a minimal computational cost in Appendix to achieve a powerful *inter-frame* representation learning.

## 4 Experiment

We evaluate our shifted chunk Transformer, denoted as SCT, on five commonly used action recognition datasets: Kinetics-400 [20], Kinetics-600 [7], Moment-in-Time [31] (Appendix), UCF101 [39] and

Table 2: Hyper-parameters of data processing and optimization.

|  | K400 | K600 | U101 | H51 | MMT |
|---|---|---|---|---|---|
| Frame Rate | 5 | 5 | 10 | 10 | 8 |
| Frame Stride | 10 | 10 | 8 | 8 | 10 |
| #Warmup epochs | 2 | 2 | 3 | 4 | 2 |
| Learning rate | 0.3 | 0.3 | 0.25 | 0.25 | 0.3 |
| Label smoothing | 0.1 | 0.1 | 0 | 0 | 0.3 |
| Dropout | 0.2 | 0.2 | 0 | 0 | 0.2 |

Table 3: The model structure of three shifted chunk Transformers.

| Model | $D$ | MLP size | $M$ | $D'$ | Clip MLP size | #Heads | #Param | GFLOPs |
|---|---|---|---|---|---|---|---|---|
| SCT-S | 96 | 384 | 4 | 192 | 768 | [4 6 8 8] | 18.72M | 88.18 |
| SCT-M | 128 | 512 | 6 | 192 | 768 | [4 8 8 8] | 33.48M | 162.90 |
| SCT-L | 192 | 768 | 4 | 192 | 768 | [4 6 8 8] | 59.89M | 342.58 |

HMDB51 [25]. We adopt ImageNet-21K for the pre-training [37, 11] because of large model capacity of SCT. The default patch size for each image token is $4 \times 4$. In the training, we use a synchronous stochastic gradient descent with momentum of $0.9$, a cosine annealing schedule [30], and the number of epochs of 50. We use batch size of 32, 16 and 8 for SCT-S, SCT-M and SCT-L, respectively. And the frame crop size is set to be $224 \times 224$. For data augmentation, we randomly select the start frame to generate the input clip. In the inference, we extract multiple views from each video and obtain the final prediction by averaging the softmax probabilistic scores from these multi-view predictions. The details of initial learning rate, optimization and data processing are shown in Table 2. All the experiments are run on 8 NVIDIA Tesla V100 32 GB GPU cards.

We construct three shifted chunk Transformers, SCT-S, SCT-M, and SCT-L, in terms of various model sizes and computation complexities. We employ four consecutive blocks with alternating one image chunk self-attention and one shifted MSA. The patch embedding size $D$, MLP dimension of these self-attentions, the number of ViLT layers $M$, the clip encoder embedding size $D'$, MLP dimension of clip encoder, the numbers of heads in ViLT, LSH attention, shifted MSA and clip encoder are shown in Table 3. Each image chunk self-attention consists of $M$ layers ViLT followed by an image LSH attention and a linear pooling layer to reduce the number of spatial dimensions. We use four-layer clip encoders to obtain the video classification results as validated in the Appendix.

**Validating frame feature extractor** We compare the frame encoder of our shifted chunk Transformer (SCT) with ViT [13] in Table 4. For a fair comparison, we only replace the ViLT with ViT, and remain all other components the same. From Table 4, we observe that 1) large models, ViT-L-16 and SCT-L, yield higher accuracy than base models, ViT-B-16 and SCT-S; 2) ViT with a small patch size achieves better accuracy than ViT with a large patch; 3) SCT-L improves the accuracy of ViT-L by 4% while using much less number of parameters and FLOPs. The tiny patch and enforced locality prior in ViLT are validated to be effective for spatio-temporal learning.

Table 4: The effect of frame extractor and the number of tokens on Top-1 accuracy (%).

| Method | Patch size | Chunk size | #Tokens | #Param | GFLOPs | Kinetics-400 |
|---|---|---|---|---|---|---|
| ViT-B-16 | 16×16 | - | 14×14 | 114.25M | 405.06 | 75.33 |
| ViT-B-16 | 12×12 | - | 18×18 | 114.25M | 665.78 | 77.12 |
| ViT-B-16 | 8×8 | - | 28×28 | 114.25M | 1603.67 | 78.95 |
| ViT-L-16 | 16×16 | - | 14×14 | 328.63M | 1413.45 | 79.15 |
| SCT-S | 4×4 | 7×7 | 8×8 | 18.72M | 88.18 | 78.41 |
| SCT-M | 4×4 | 7×7 | 8×8 | 33.48M | 162.90 | 81.26 |
| SCT-L | 4×4 | 7×7 | 8×8 | 59.89M | 342.58 | 83.02 |

Table 5: The effect of the number of shifted MSA layers and shifted frames on Top-1 accuracy (%).

| Method | #Shifted MSA | #Shifted frame | K400 | U101 |
|--------|:---:|:---:|:---:|:---:|
| SCT-S | 0 | 0 | 76.91 | 97.01 |
| SCT-S | 1 | 1 | **78.41** | **98.02** |
| SCT-S | 1 | 5 | 77.02 | 97.15 |
| SCT-S | 2 | 1 | 77.45 | 97.33 |

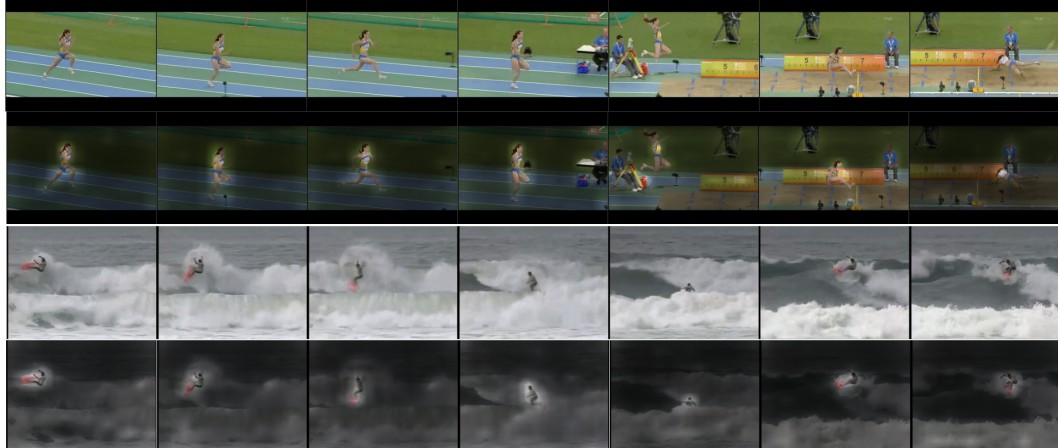

Figure 4: Visualization of the patch and frame attention maps (the second and fourth rows).

**Effect of shifted MSA**  We conduct experiments on Kinectis-400 and UCF101 to validate the hyper-parameters of the shifted MSA layer, including the number of shifted MSA layers, and the number of shifted frames used in the calculation of key in equation (6). All other network configurations follow the Table 3. The #Shifted MSA of 0 and #Shifted frame of 0 in Table 5 mean that one standard MSA is used instead of shifted MSA. From Table 5, we observe that 1) a shifted MSA improves the accuracy up to 1.5% compared with the conventional MSA; 2) one layer shifted MSA with the shifted number of frames of one yields the best accuracy. The shifted MSA explicitly formulates the motion effect of object by considering the nearby frames, which improves the accuracy for video classification. We use one layer shifted MSA with the number of shifted frames of one in our experiment.

**Varying the number of input frames and temporal views**  In our experiments so far, we have kept the number of input frames fixed to 24 across different datasets. To discuss the effect of the number of input frames on video level inference accuracy, we validate the number of input frames of 24, 48, 96, and the number of temporal views from 1 to 8. Fig. 3 shows that as we increase the number of frames, the accuracy using a single clip increases, since the network is incorporated longer temporal information. However, as the number of used views increases, the accuracy difference is reduced. We use the number of frames of 24 and the number of temporal views of 4 in our experiment.

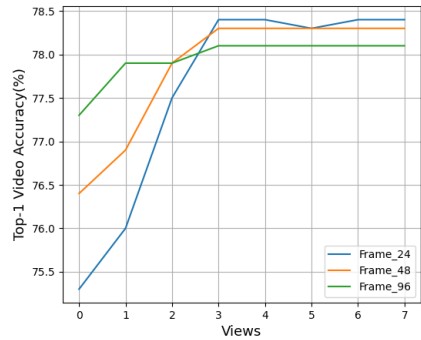

Figure 3: The effect of varying the number of input frames and temporal views.

**Patch and frame attention**  Our shifted chunk Transformer (SCT) can detect fine-grained discriminative regions for each frame in the entire clip in Fig. 4. Specifically, we average attention weights of the shifted MSA across all heads and then recursively multiply the weight matrices of all layers [1], which accounts for the attentions through all layers. The designed framework of SCT leads to an easy diagnosis and explanation for

Table 6: Top-1 and Top-5 accuracy (%) comparisons to state-of-the-art approaches on Kinectics-400.

| Method | TFLOPs×Views | #Param | Runtime (s) | Top1 | Top5 |
|---|---|---|---|---|---|
| TEA [27] | 0.07×10×3 | - | - | 76.1 | 92.5 |
| I3D NL [49] | - | 54M | - | 77.7 | 93.3 |
| CorrNet-101 [46] | 0.224×10×3 | - | - | 79.2 | - |
| ip-CSN-152 [42] | 0.109×10×3 | 33M | - | 79.2 | 93.8 |
| SlowFast [16] | 0.234×10×3 | 60M | - | 79.8 | 93.9 |
| X3D-XXL [15] | 0.194×10×3 | 20M | 0.176 | 80.4 | 94.6 |
| TimeSformer [5] | 2.38×1×3 | 121M | 0.475 | 80.7 | 94.7 |
| MViT-B 64×3 [14] | 0.455×3×3 | 37M | 0.153 | 81.2 | 95.1 |
| ViViT-L [4] | 399.2×4×3 | 89M | - | 81.3 | 94.7 |
| SCT-S | 0.088×4×3 | 19M | 0.051 | 78.4 | 93.8 |
| SCT-M | 0.163×4×3 | 33M | 0.072 | 81.3 | 94.5 |
| SCT-L | 0.343×4×3 | 60M | 0.106 | **83.0** | **95.4** |

Table 7: Classification accuracy (%) comparisons to state-of-the-art approaches on Kinectics-600, UCF101 (3 splits) and HMDB51 (3 splits). 'K400' denotes pretraining on Kinetics-400 and ImageNet. 'K600' denotes pretraining on Kinetics-600 and ImageNet.

| Method | Views | #Para | Top1 | Top5 |
|---|---|---|---|---|
| AttenNAS [48] | - | - | 79.8 | 94.4 |
| LGD-3D [35] | - | - | 81.5 | 95.6 |
| SlowFast [16] | 10×3 | 60M | 81.8 | 95.1 |
| X3D-XL [15] | 10×3 | 11M | 81.9 | 95.5 |
| TimeSformer [5] | 1×3 | 121M | 82.4 | 96.0 |
| ViViT-L [4] | 4×3 | 89M | 83.0 | 95.7 |
| SCT-S | 4×3 | 19M | 77.5 | 93.1 |
| SCT-M | 4×3 | 33M | 81.7 | 95.5 |
| SCT-L | 4×3 | 60M | **84.3** | **96.3** |

| Method | Pretrain | U101 | H51 |
|---|---|---|---|
| I3D [8] | K400 | 95.4 | 74.5 |
| ResNeXt [17] | K400 | 94.5 | 70.2 |
| R(2+1)D [43] | K400 | 96.8 | 74.5 |
| S3D-G [51] | K400 | 96.8 | 75.9 |
| LGD-3D [35] | K600 | 97.0 | 75.7 |
| SCT-S | ImageNet | **98.0** | 76.5 |
| SCT-L | ImageNet | 97.7 | **81.4** |
| SCT-S | K400 | 98.3 | 81.5 |
| SCT-M | K400 | 98.5 | 83.2 |
| SCT-L | K400 | **98.7** | **84.6** |

the prediction, which potentially makes SCT applicable to various critical fields, *e.g.*, healthcare and autonomous driving.

**Comparison to state-of-the-art approaches** We compare our shifted chunk Transformer (SCT) to the current state-of-the-art approaches based on the best hyper-parameters validated in the previous ablation studies. We obtain the results of previous state-of-the-art approaches from their papers. We obtain the actual runtime (s) in one single NVIDIA V100 16GB GPU by averaging 50 inferences with batch size of one. In Kinetics-400 and Kinetics-600, we initialize our ViLT and LSH attention trained on ImageNet-21K.

Our shifted chunk Transformers (SCT) surpass previous state-of-the-art approaches including both recent Transformer based video classification and previous deep ConvNets based methods by 2.7%, 1.3%, 1.7% and 8.9% on Kinectis-400, Kinectic-600, UCF101 and HMDB51 in Table 6-7 based on RGB frames, respectively. The local scheme and LSH approximation in image chunk self-attention enables to use patches of a small size. Because of the efficient model design, SCT achieves the best accuracy even only using pretraining on the ImageNet on UCF101 and HMDB51. Besides the higher action recognition accuracy, the SCT employs less number of parameters and FLOPs than ViViT because we employ less number of channels and our SCT is effective for spatio-temporal learning.

## 5   Conclusion

In this work, we proposed a new spatio-temporal learning called shifted chunk Transformer inspired by the recent success of vision Transformer in image classification. However, the current pure-Transformer based spatio-temporal learning is limited by computational efficiency and feature

robustness. To address these challenges, we propose several efficient and powerful components for spatio-temporal Transformer, which is able to learn fine-grained features from a tiny image patch and model complicated spatio-temporal dependencies. We construct an image chunk self-attention which leverages locality-sensitive hashing to efficiently capture fine-grained local representation with a relatively low computation cost. Our shifted self-attention can effectively model complicated *inter-frame* variances. Furthermore, we build a clip encoder based on pure-Transformer for frame-wise attention and long-term *inter-frame* dependency modeling. We conduct thorough ablation studies to validate each component and hyper-parameters in our shifted chunk Transformer. It outperforms previous state-of-the-art approaches including both pure-Transformer architectures and deep 3D convolutional networks on various datasets in terms of accuracy and efficiency.

## 6 Acknowledgement

This work is supported by Kuaishou Technology. No external funding was received for this work. Moreover, we would like to thank Hang Shang for insightful discussions.

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
