# Shifted Chunk Transformer for Spatio-Temporal Representational Learning

**Xuefan Zha**
Kuaishou Technology
zhaxuefan@kuaishou.com

**Wentao Zhu**
Kuaishou Technology
wentaozhu@kuaishou.com

**Tingxun Lv**
Kuaishou Technology
lvtingxun@kuaishou.com

**Sen Yang**
Kuaishou Technology
senyang@kuaishou.com

**Ji Liu**
Kuaishou Technology
ji.liu.uwisc@Gmail.com

## A  Appendix

**Varying the number of clip encoder layers**   See Table 1. From the table, adding two more layers of clip encoder only increases the number of parameters by 1M. The clip encoder using four layers yields better accuracy. We use four layered clip encoder in our experiment.

Table 1: Effect of clip encoder on the Top-1 accuracy (%)

| Name | #Layer | #Param | FLOPs | K400 | U101 |
|------|--------|--------|-------|------|------|
| SCT-S | 2 | 17.67M | 86.14G | 76.32 | 97.35 |
| SCT-S | 4 | 18.72M | 88.18G | **78.41** | **98.02** |

**Pretrained model analysis**   See Table 2. Pretraining on large amount of data yields better top-1 accuracy.

Table 2: Pretrain Model Result

| Name | Pretrain Type | Top-1 Acc |
|------|---------------|-----------|
| **UCF101** | | |
| SCT-S | ImageNet | 98.02% |
| SCT-M | ImageNet | 97.45 % |
| SCT-L | ImageNet | 97.70 % |
| SCT-S | ImageNet+Kinetics-400 | 98.33 % |
| SCT-M | ImageNet+Kinetics-400 | 98.45 % |
| SCT-L | ImageNet+Kinetics-400 | **98.71** % |
| **HMDB51** | | |
| SCT-S | ImageNet | 76.52 % |
| SCT-M | ImageNet | 78.31 % |
| SCT-L | ImageNet | 81.42 % |
| SCT-S | ImageNet+Kinetics-400 | 81.54 % |
| SCT-M | ImageNet+Kinetics-400 | 83.22 % |
| SCT-L | ImageNet+Kinetics-400 | **84.61** % |

35th Conference on Neural Information Processing Systems (NeurIPS 2021).

**Results on Moments in Time [10]** See Table 3. We achieve comparable accuracy with much less number of parameters and GFLOPs on Moments in Time [10].

Table 3: Results on Moments in Time [10]

| Method | #Param | GFLOPs | Top-1 (%) | Top-5 (%) |
|---|---|---|---|---|
| I3D [4] | 25M | - | 29.5 | 56.1 |
| ViViT-L [2] | 88.9M | 1446 | 38.0 | 64.9 |
| VATT-L [1] | 306.1M | 29800 | 41.1 | 67.7 |
| SCT-M | 33.48M | 162.9 | 36.8 | 61.2 |
| SCT-L | 59.89M | 342.6 | 37.3 | 65.1 |

**Ablation study on ViLT** We further compare the ViLT with convolution variants and one Transformer variant, i.e., LSH attention. We compare ViLT (78.4%, 98.3%) with various convolution variants in SCT-S on the Kinetics-400 and UCF101 datasets. We have convolution (73.9%, 94.9%), convolution + bn (74.3%, 95.0%), and residual convolution block (75.1%, 95.8%), which sufficiently demonstrates the effectiveness of our ViLT. From the perspective of receptive field size, without pooling, a four layered ConvNet with 3x3 kernel has receptive field size of 9x9, and our ViLT is able to fully model the information from 28x28 of each chunk. Replacing ViLT with image LSH attention obtains 76.6% and 96.1% Top-1 accuracy, because the LSH self-attention reduces the computation by approximating the dense matrix with an upper triangular matrix.

**Ablation study on image LSH attention** To conduct ablation studies for image LSH attention, we a) remove the ViLT and obtain 63.2% and 85.2% Top-1 accuracy on Kinetics-400 and UCF101, because it fails to capture low-level fine grained features, b) replace LSH attention with ConvNets and obtain convolution (75.3%, 96.6%), convolution + bn (75.6%, 96.9%), and residual convolution block (76.9%, 97.0%), because ConvNets have limited receptive field size compared with Transformers, c) remove the LSH attention in SCT-S and achieve (76.2%, 96.5%) Top-1 accuracy. The global attention brought by LSH attention in each frame helps spatio-temporal learning.

**Ablation study on shifted MSA** Compared with the conventional self-attention only modeling the intra-frame patches (space attention), or divided space-time attention only modeling the same position along different frames which cannot handle big motions, our shifted self-attention explicitly models the motion and focuses the main objects in the video. We also validate the effectiveness of our shifted attention through ablation study and comparison with previous state-of-the-art methods.

Empirically, we compare the shifted MSA with various attentions, i.e., space attention (conventional self-attention, 77.02%), time attention [3] (77.62%), and concatenated feature from space and time attentions [3] (77.35%) with fixed other components in SCT-S on the Kinetics-400 dataset, which demonstrates the advantages of explicitly effective motion modeling in the shifted attention. The attention map visualization in Fig. **??** also verifies the effective motion capture of the main object in the video.

**Results on SSv2 and Diving-48** We further conduct experiments on Something-Something-V2 [7] and Diving-48 [9], which are more dynamic datasets and rely heavily on the temporal dimension. Our SCT-L with Kinetics-600 pretrained model obtain 68.1% and 81.9% accuracy on the two datasets, respectively, compared with TEA [8] (65.1%, N/A), SlowFast [6] (61.7%, 77.6%), ViViT-L/16x2 [2] (65.4%, N/A), TimeSformer-L [3] (62.4%, 81.0%), and MViT-B, 32x3 [5] (67.8%, N/A). Our SCT-L achieves the best Top-1 accuracy on the two datasets.

**Hyper-parameters of shifted MSA** In our experiment, the frame rate of each input clip is varied from 5-10, which is 0.1-0.3s. From the perspective of human vision system, the typical duration of persistence of vision is 0.1-0.4s. The experiment validates the best numbers of shifted MSA and shifted frames are 1, which is consistent with our vision system and the bigger number of shifted frames could misses the motion information for some actions. From the perspective of model complexity, we have the multi-layer clip encoder after shifted MSA to specifically model complicated

inter-frame dependencies. The shifted MSA is forced to learn fine-grained motion information. In the future work, developing multi-scale shifted MSA is an interesting topic.