# OpenReview forum: "Shifted Chunk Transformer for Spatio-Temporal Representational Learning"
_NeurIPS.cc/2021/Conference — NeurIPS 2021 Poster_

### Official Review · Reviewer_DdYC · 2021-07-09

**Rating:** 6
**Confidence:** 4

**Summary:**

The paper proposes a new transformer-based architecture for video classification. The proposed architecture builds on the recently introduced vision transformer model (ViT), which first decomposes a given image/video frame into a set of non-overlapping patches. However, unlike in prior work,  instead of applying self-attention directly on patches, the authors group locally adjacent patches into local "chunks", and apply self-attention on top of these chunks. To ensure that the model can propagate local chunk information globally across the entire image/video frame, the authors introduce  shifted self-attention. Lastly, a separate clip encoder is used to incorporate temporal video information. The proposed approach achieves state-of-the-art on several action recognition benchmarks such as Kinetics-400, and Kinetics-600.

**Limitations And Societal Impact:**

Limitations of the proposed approach are not discussed, i.e., Section 5 only contains limitations for previous work.

**Main Review:**

Strengths:
- Video transformers are growing in popularity. Therefore, the proposed approach that addresses several limitations of prior video transformer based methods is timely and relevant.
- The proposed approach achieves impressive results on Kinetics-400, and Kinetics-600.
- The proposed method is also more efficient than prior transformer based techniques.

Weaknesses:
- Personally, I found the draft quite difficult to understand due to the writing related issues (i.e., incoherent or grammatically incorrect sentences), missing technical details, and the technical presentation style. Below I discuss some of these issues in more detail.
- The paragraph on "Image locality-sensitive hashing (LSH) attention" is confusing and difficult to understand both due to numerous writing issues, and also due to the lack of details related to LSH. The authors talk about "bucketing" but it would be useful to include a brief technical overview so that the readers who are less familiar with LSH would still understand the approach without having to look up other papers.
- The description of the shifted attention could be significantly improved. Could the authors make a figure illustrating this concept and include it in the draft? Unlike written in the draft, Figure 1 doesn't illustrate how shifted attention mechanism works in practice. To be honest, I'm still confused about it. From Equation 6, it seems that the key features are computed from time t-1 and that there is no temporal overlap between query and value features, both of which are computed at time t. Therefore, I don't see how the temporal information is propagated into the feature representation y' (as in Eq.3 ), since the key features are only used to compute similarity matrix, but not the new feature representation y' . Isn't that a limiting factor in the model design? In my view, a new feature representation y' that incorporates information from both time t-1, and also time t would be more powerful. What is the motivation of such model design? Am I missing something? Having said this, the results in Table 5 indicate that the proposed scheme leads to ~1.5% improvement, which is quite substantial.
- The authors make claims about their approach learning inter-frame motion of objects. However, Kinetics, HMDB, and UCF are known to be heavily biased towards spatial scene dynamics. To verify these claims, one would need to conduct experiments on temporally heavy datasets such as Something-Something-V2 and Diving-48. I would urge the authors to conduct these experiments for the rebuttal.
- Many relevant experimental details are missing or they are inconsistent with the previously published work. For example, in Table 1, the authors list that ViT-B achieves 75.3 % accuracy on Kinetics-400. How's this baseline implemented? For example, in the TimeSformer paper [4] , the accuracy of a 8-frame spatial-only model is listed as 76.9 % on Kinetics-400. Therefore, I was wondering what's the reason behind this gap. Furthermore, in Table 4, the authors list ViT-B to have 196 tokens, and ~405 GFLOPs. How are these numbers computed? The 8-frame spatial-only ViT-B has the cost of ~141 GFLOPs. How many frames are all of these models using? These details should be included both in the paper, and in the tables.
- The term "image local chunk self-attention", which is used many times throughout the paper is cumbersome. I would urge the authors to rename their model to something more elegant.
- The paper often feels like code/implementation documentation. The authors tend to focus on very low-level implementation details too much (e.g., Sections 3.1-3.4), making the paper more difficult to read.

Questions / Comments:
- For pretraining, do the authors pretrain their model on ImageNet-21k themselves, or do they use publicly available ImageNet-21k pretrained models? If it's the former, it would be good to include pretraining related details. If it's the latter, for reproducibility, it would be good to know, which public models the authors are using, as the models displayed in Table 3 have a custom design (compared to standard ViT).
- Could the authors also include the total number of TFLOPs in their tables so that it would be easier to compare computational complexity of each method?
- In Table 5, the performance of using two layers of shifted MSA drops quite substantially, which is a bit counterintuitive. Do the authors have an explanation for this?
- In Table 4, what exactly does the chunk size (i.e., 49) mean? Does it mean that it encompasses 49 patches on each axis? It seems that this should be 7 instead of 49.
- All the tables list the # of tokens for a single frame. However, I assume the model operates on multiple frames. Therefore, it would be useful to include the number of used frames in the tables.

**Time Spent Reviewing:**

4 hours

---

> ### Author Response · Authors · 2021-08-10
> **Response to Reviewer DdYC (part 2/2): Technical details for question/comments.**
>
> Thank you for your question! We include more technical details as below.
>
> * Pretrained model
>
> In the UCF101 and HMDB51 dataset, we have provided pre-training model analysis in Appendix A. In Kinetics-400 and Kinetics-600, we initialize our image local chunk attention and LSH attention trained on ImageNet-21K.
>
> * TFLOPs
>
> We will add TFLOPs in the Table 6. We have GFLOPs in Table 6, and 1 TFLOPs is equivalent to 1,000 GFLOPs.
>
> * Hyper-parameters of shifted MSA
>
> In our experiment, the frame rate of each input clip is varied from 5-10, which is 0.1-0.3s. From the perspective of human vision system, the typical duration of persistence of vision is 0.1-0.4s. The experiment validates the best numbers of shifted MSA and shifted frames are 1, which is consistent with our vision system and the bigger number of shifted frames could misses the motion information for some actions. From the perspective of model complexity, we have the multi-layer clip encoder after shifted MSA to specifically model complicated inter-frame dependencies. The shifted MSA is forced to learn fine-grained motion information. In the future work, developing multi-scale shifted MSA is an interesting topic.
>
> * Chunk size
>
> Chunk size is the number of local patches in each image local chunk self-attention. The chunk size of 49 comes from $7\times7$ patches within a chunk. The tiny patch size used is $4\times4$, and each image local chunk self-attention takes a window of size $28\times28$ as the input without the linear pooling layer. Each frame of size $224\times224$ can be split into $64=8\times8$ chunks, which is the number of Tokens defined. The ViT [12] has no image local Transformer, and the number of Tokens is directly obtained from $(224/Patch size)^2$.
>
> * The number of frames
>
> The number of input frames is fixed to 24 across different datasets in Line 299.
>
> ## Reference
> [12] Dosovitskiy et al. An image is worth 16x16 words: Transformers for image recognition at scale. ICLR, 2021.

---

> > ### Comment · Reviewer_DdYC · 2021-08-26
> > **Post-rebuttal Feedback**
> >
> > The rebuttal addresses many of my concerns. Therefore, I'm increasing my original rating. I would urge the authors to improve the paper's presentation (both writing and figures), and also include the additional experiments.

---

> > > ### Author Response · Authors · 2021-09-02
> > > **Response to Reviewer DdYC: Thank you for your suggestions!**
> > >
> > > Thanks a lot for your suggestions! All feedbacks from reviewers lead to improvements of this paper. We will improve the paper’s presentation accordingly. Additional experiments will be added in the manuscript.
> > >
> > > Code and trained models are being released soon.

---

> ### Author Response · Authors · 2021-08-10
> **Response to Reviewer DdYC (part 1/2): more results and analysis**
>
> Thank you for your question!
>
> * A brief technical overview of LSH
>
> The problem of finding nearest neighbors quickly in high-dimensional spaces can be solved by locality-sensitive hashing (LSH), which hashes similar input items into the same "buckets" with high probability.
>
> LSH attention [20], which is firstly proposed in NLP to deal with prohibitively costly training large Transformers on long sequences, replaces dot-product attention with one that uses locality-sensitive hashing (LSH), changing its complexity from $O (L^2)$ to $O (LlogL)$, where $L$ is the length of the sequence. Inspired by this, we develop image LSH attention block and integrate the image LSH attention into our framework to learn from massive tiny patches. We will add the overview of LSH in section 3.2.
>
> * The Analysis of Shift MSA
>
> We draw a figure to illustrate the shifted MSA in https://anonymous.4open.science/r/Paper-explanation-image-2414/img-2.png
>
> To incorporate information from the current time t, the image chunk self-attention block is adequate to learn effective intra-frame features. The shifted MSA block is assumed to able to extract fine-grained motion information along two frames.
>
> In Transformer-based temporal modeling, TimeSformer [4] conducts a systematic investigation on various attentions, e.g., space attention, joint space-time attention which is a full Transformer along both space and time, divided space-time attention which employs the current frame and the same position along time domain, sparse local global attention, and axial attention. TimeSformer shows that the joint space-time attention achieves worse accuracy than divided space-time attention, which can be considered as a factorized space-time attention, because the joint space-time attention consumes too much memory and divided space-time attention is much more efficient and able to learn from a big space size and a long sequence.
>
> In our work, we conduct an in-depth design of the internal structure in the attention by employing the $t-\Delta t$ frame to construct the key matrix. In this way, we do not need to sacrifice the space size along the time dimension as in the divided space-time attention. And the in-depth modification does not increase extra memory or FLOPs although we can only model two frames. The design of employing previous frame information to instruct feature learning/extraction in the current frame enables motion guided on-line sequence learning and inference.
>
> Specifically, the dot-product $\text{softmax}(Q\_t K^T\_t / \sqrt{d\_k})$ in the self-attention [42] yields a similarity matrix
> $S\_t = (s\_{(t,i),(t,j)})\_{i=1,...,L, j=1,...,L}$. Let $V\_t$ be $\[v\_{(t,1)}; v\_{(t,2)}; ...; v\_{(t,L)}\]$.
> The conventional self-attention [42] yields feature
> $(\text{Attention} (Q\_t, K\_t, V\_t))\_{i, j} = {\sum}\_k { s\_{(t,i),(t,k)} v\_{t, k, j} }$.
>
> In our shifted MSA, because we employ the $t - \Delta t$ frame to construct the key matrix, the similarity matrix becomes
> $S\_t = (s\_{(t,i),(t-\Delta t,j)})\_{i=1,...L, j=1,...L}$. If $s\_{(t,i),(t-\Delta t,j)}$ is bigger, it means the patch $i$ in the $t$-th frame is similar with the patch $j$ in the ($t-\Delta t$)-th frame. In other words, it is more likely that, the patch $j$ moves to patch $i$. If $s\_{(t,i),(t-\Delta t,j)}$ is smaller, it means it is more likely a noisy background patch, not the main object. Then the shifted self-attention yields feature
> $( \text{ShiftAttention} (Q\_t, K\_t, V\_t) )\_{i, j} = {\sum}\_k {s\_{(t,i),(t-\Delta t,k)} v\_{t, k, j} }$, which means that we emphasize the main object and the patch moved from the previous $t-\Delta t$ frame.
>
> Compared with the conventional self-attention only modeling the intra-frame patches, or divided space-time attention only modeling the same position along different frames which cannot handle big motions, our shifted self-attention explicitly models the motion and focuses the main objects in the video. We also validate the effectiveness of our shifted attention through ablation study and comparison with previous state-of-the-art methods.
>
> Empirically, we compare the shifted MSA with various attentions, i.e., space attention (conventional self-attention, 77.02%), time attention [4] (77.62%), and concatenated feature from space and time attentions (incorporating information from both time t-1 and time t, 77.35%) with fixed other components in SCT-S on the Kinetics-400 dataset, which demonstrates the advantages of explicitly effective motion modeling in the shifted attention. The attention map visualization in Fig. 3 also verifies the effective motion capture of the main object in the video.
>
>
> * More experimental results on SSv2 and Diving-48
>
> We further conduct experiments on SSv2 and Diving-48, which are more dynamic datasets and rely heavily on the temporal dimension. Our SCT-L with Kinetics-600 pretrained model obtain 68.1% and 81.9% accuracy on the two datasets, respectively, compared with TEA [25] (65.1%, N/A), SlowFast [15] (61.7%, 77.6%), ViViT-L/16x2 [3] (65.4%, N/A), TimeSformer-L [4] (62.4%, 81.0%), and MViT-B, 32x3 [13] (67.8%, N/A). Our SCT-L achieves the best Top-1 accuracy on the two datasets. We will add VATT and update Table 3 in the supplementary.
>
> * Gap of relevant experimental details
>
> 1) In Table 1, it comes from ViT-B with a four-layer clip encoder to show our motivation of using small patch size.
> 2) This accuracy gap might come from different pretrain models and other experimental variance. We do not use any pretrained model in Table 1 because it is only used to show the motivation. Instead, TimeSformer [4] employs ImageNet-21K for the pretraining.
> 3) In Table 4, the ViT [12] treats each image patch as one token. The input image size is $224\times224$, and the patch size is $16\times16$, which is the same as TimeSformer [4]. The ViT has no image local Transformer, and the number of Tokens is directly obtained from $(224/16)^2 = 196$.
> 4) We use the number of frames of 24 and the number of temporal views of 4 in our experiment for the 8-frame spatial-only ViT-B.
>
> * Rename problem
>
> We will add these details into the revision and update the "image local chunk self-attention" with "visual local transformer (ViLT)".
>
> * Too much low-level implementation details
>
> We will add figures and motivations for the shifted MSA and make the paper easy to follow.
>
>
> ## Reference
>
> [3] Anurag Arnab, Mostafa Dehghani, Georg Heigold, Chen Sun, Mario Luciˇ c, and Cordelia ´ Schmid. Vivit: A video vision transformer. arXiv preprint arXiv:2103.15691, 2021. \
> [4] Gedas Bertasius, Heng Wang, and Lorenzo Torresani. Is
> space-time attention all you need for video understanding? ICML, 2021. \
> [12] Dosovitskiy et al. An image is worth 16x16 words: Transformers for image recognition at scale. ICLR, 2021. \
> [13] Haoqi Fan, Bo Xiong, Karttikeya Mangalam, Yanghao Li, Zhicheng Yan, Jitendra Malik, and Christoph Feichtenhofer. Multiscale vision transformers. arXiv preprint arXiv:2104.11227, 2021. \
> [15] Christoph Feichtenhofer, Haoqi Fan, Jitendra Malik, and Kaiming He. Slowfast networks for video recognition. In Proceedings of the IEEE/CVF International Conference on Computer Vision, pages 6202–6211, 2019. \
> [20] Kitaev et al. Reformer: The efficient transformer. ICLR, 2020. \
> [25] Yan Li, Bin Ji, Xintian Shi, Jianguo Zhang, Bin Kang, and Limin Wang. Tea: Temporal excitation and aggregation for action recognition. In Proceedings of the IEEE/CVF Conference on Computer Vision and Pattern Recognition, pages 909–918, 2020. \
> [42] Vaswani et al. Attention is all you need. In Advances in Neural Informa469 tion Processing Systems, 2017.

---

### Official Review · Reviewer_yF1i · 2021-07-16

**Rating:** 6
**Confidence:** 4

**Summary:**

The paper proposes a novel transformer-based architecture for video processing. The input tokens are small patches extracted for each frame. The model consists of 4 modules: First, the tokens are processed locally, using a transformer with restricted, local connectivity (each image is split into multiple chunks and only the patches from the same chunk exchange messages using the self-attention layer). Second, a global processing is conducted using an efficient transformer (that use LSH attention approximation) to capture long-range interactions. In order to take into account the motion effect, a shifted self-attention is introduced, modifying the classical self-attention layer to use keys computed from the previous time step. Similar to language modelling, a special token ([CLS]) is used to extract frame-level representations that are fed into a final transformer block to compute the final video representation.

**Limitations And Societal Impact:**

The limitations are not addressed in the paper.

**Main Review:**

*Pros:*
- Transformers have been already successfully applied in image processing. However, fewer methods managed to extend them to the video domain.
- The proposed method is more efficient (in terms of FLOPS and params) than other transformers used for video processing, as shown in experiments.
- The experiments show improvement over existing methods, proving that the architecture is effective.

*Cons:*
 - Even if efficient transformers for spatio-temporal data represents a useful direction for the computer-vision community, my main concern regards the novelty brought by this work and the relevance for the NeurIPS community. The model shows how to combine/modify different existing models but fails to clearly validate the usefulness of each individual one for the final performance. The paper contains a set of ablations but, in my opinion, the central contributions are still not empirically validated as I will explained in the following:

- The frame-level processing consisting of an “Image Local Chunk Attention” and an “Image LSH Attention” is only compared against a ViT model. Some questions regarding this approach are still open: Is it necessary to have a local processing (the chunk-level one) besides the global one? Is this local processing better than a convolutional block, since both approaches aim to process local information? Is the global processing (LSH Attention) necessary? I would suggest to include more ablation studies by a)dropping the Image Local Chunk Attention; b) replacing it with a convolutional network; c) dropping the LSH Attention

- The “Image Local Chunk Attention” was designed to process the information locally. The same motivation and a very similar solution is proposed in [34]. A discussion regarding the similarities/differences but more importantly the possible advantages would be appreciated.

- The idea of shifting the temporal dimension to capture the objects’ motion efficiently is interesting (and as far as I know novel in transformer literature) and seems to be effective. However the paper lacks an intuitive motivation for this. What does it mean to aggregate global information based on the similarities computed w.r.t previous time steps?

- To fully validate the usefulness of the proposed model in video datasets, results on more dynamic datasets, where the temporal dimension plays a crucial role such as Smt-Smt V2 would be appreciated. The authors  provide some initial results on the Moments in Time datasets, but the comparison doesn’t seem to be beneficial. Moreover, important comparisons are missing from Table 3 in supplementary (e.g. VATT method that obtain better results)


Overall I believe that, with proper validations in terms of additional ablation studies, this work could be relevant for the  CV community. However, I don’t think that the novelty brought by this work is sufficient for the acceptance at NeurIPS.

######## POST-REBUTTAL #########

I appreciate the additional experiments included in the rebuttal. In my opinion, they validate each component and gives me more confidence in the proposed method. I am still not fully convinced by the explanation for how the shifted module models temporal information, but the quantitative results are quite impressive, especially on heavy temporal datasets (such as SSv2 and Diving-48 included in the rebuttal).

I will increase my score to 6: Marginally above the acceptance threshold, but I strongly encourage the authors to include all the ablation studies in the manuscript, as neither those experiments nor similar ones exist in the current version of the paper, and as other reviewers suggested, it is very important to understand the benefits of using each component of the model.

**Time Spent Reviewing:**

7

---

> ### Author Response · Authors · 2021-08-10
> **Response to Reviewer yF1i: more ablations and analysis.**
>
> With our due respect for the reviewer’s time, we humbly yet firmly suggest that the reviewer might have an important misunderstanding or underestimation of this paper’s true merit.\
> We start by strongly pointing out: “combine/modify different existing models but fails to clearly validate the usefulness of each individual one for the final performance.” is an incorrect assessment of our main points. We quote Reviewer zDkv who precisely summarized our contributions: “The paper has addressed several painpoints of video spatio-temporal modeling using transformers. The authors have conducted thorough ablation to dissect their approach”.
>
> * Image local chunk self-attention is necessary.
>
> We further compare the image local chunk self-attention with convolution variants and one Transformer variant, i.e., LSH attention. We compare image local chunk self-attention (78.4%, 98.3%) with various convolution variants in SCT-S on the Kinetics-400 and UCF101 datasets. We have convolution (73.9%, 94.9%), convolution + bn (74.3%, 95.0%), and residual convolution block (75.1%, 95.8%), which sufficiently demonstrates the effectiveness of our image local chunk self-attention. From the perspective of receptive field size, without pooling, a four layered ConvNet with 3x3 kernel has receptive field size of 9x9, and our local chunk self-attention is able to fully model the information from 28x28 of each chunk. \
> Replacing image local chunk self-attention with image LSH attention obtains 76.6% and 96.1% Top-1 accuracy, because the LSH self-attention reduces the computation by approximating the dense matrix with an upper triangular matrix.
>
>
> * Image LSH attention is necessary.
>
> To conduct ablation studies for image LSH attention, we a) remove the image local chunk attention and obtain 63.2% and 85.2% Top-1 accuracy on Kinectics-400 and UCF101, because it fails to capture low-level fine grained features, b) replace LSH attention with ConvNets and obtain convolution (75.3%, 96.6%), convolution + bn (75.6%, 96.9%), and residual convolution block (76.9%, 97.0%), because ConvNets have limited receptive field size compared with Transformers, c) remove the LSH attention in SCT-S and achieve (76.2%, 96.5%) Top-1 accuracy. The global attention brought by LSH attention in each frame helps spatio-temporal learning.
>
>
> * Discussion of [34] and our image local chunk attention
>
> Our image local chunk attention stands in the middle of [34] and ViT [12]. [34] employs each raw pixel as the input and has a local window size of $3\times3$ inspired by convolution. Our image local chunk attention firstly employs a linear patch embedding matrix $E$ to extract feature from a tiny patch of size $4\times4$. Then we build a self-attention in a chunk of size $7\times7$ tiny patches, which means one single image local chunk attention has a window size of $28\times28$. ViT [12] instead constructs self-attention across the whole image and employs a much bigger input patch size. The advantage of image local chunk attention over ViT [12] has been demonstrated in Table 4. Our image local chunk attention has a much bigger window size compared with [34], and a linear patch embedding is sufficient to learn from a $4\times4$ tiny patch. The main difference is visualized as: https://anonymous.4open.science/r/Paper-explanation-image-2414/img-1.png
>
> * Motivation of shifted MSA
>
> Temporal modeling plays a critical role for video classification. Methods have been explored in previous work including optical flow [7], 3D convolution [14], joint/divided space-time attention [4], etc.
>
> In Transformer-based temporal modeling, TimeSformer [4] conducts a systematic investigation on various attentions, e.g., space attention, joint space-time attention which is a full Transformer along both space and time, divided space-time attention which employs the current frame and the same position along time domain, sparse local global attention, and axial attention. TimeSformer shows that the joint space-time attention achieves worse accuracy than divided space-time attention, which can be considered as a factorized space-time attention, because the joint space-time attention consumes too much memory and divided space-time attention is much more efficient and able to learn from a big space size and a long sequence.
>
> In our work, we conduct an in-depth design of the internal structure in the attention by employing the $t-\Delta t$ frame to construct the key matrix. In this way, we do not need to sacrifice the space size along the time dimension as in the divided space-time attention. And the in-depth modification does not increase extra memory or FLOPs although we can only model two frames. The design of employing previous frame information to instruct feature learning/extraction in the current frame enables motion guided on-line sequence learning and inference.
>
> Specifically, the dot-product $\text{softmax}(Q\_t K^T\_t / \sqrt{d\_k})$ in the self-attention [42] yields a similarity matrix
> $S\_t = (s\_{(t,i),(t,j)})\_{i=1,...,L, j=1,...,L}$. Let $V\_t$ be $\[v\_{(t,1)}; v\_{(t,2)}; ...; v\_{(t,L)}\]$.
> The conventional self-attention [42] yields feature
> $(\text{Attention} (Q\_t, K\_t, V\_t))\_{i, j} = {\sum}\_k { s\_{(t,i),(t,k)} v\_{t, k, j} }$.
>
> In our shifted MSA, because we employ the $t - \Delta t$ frame to construct the key matrix, the similarity matrix becomes
> $S\_t = (s\_{(t,i),(t-\Delta t,j)})\_{i=1,...L, j=1,...L}$. If $s\_{(t,i),(t-\Delta t,j)}$ is bigger, it means the patch $i$ in the $t$-th frame is similar with the patch $j$ in the ($t-\Delta t$)-th frame. In other words, it is more likely that, the patch $j$ moves to patch $i$. If $s\_{(t,i),(t-\Delta t,j)}$ is smaller, it means it is more likely a noisy background patch, not the main object. Then the shifted self-attention yields feature
> $( \text{ShiftAttention} (Q\_t, K\_t, V\_t) )\_{i, j} = {\sum}\_k {s\_{(t,i),(t-\Delta t,k)} v\_{t, k, j} }$, which means that we emphasize the main object and the patch moved from the previous $t-\Delta t$ frame.
>
> Compared with the conventional self-attention only modeling the intra-frame patches (space attention), or divided space-time attention only modeling the same position along different frames which cannot handle big motions, our shifted self-attention explicitly models the motion and focuses the main objects in the video. We also validate the effectiveness of our shifted attention through ablation study and comparison with previous state-of-the-art methods.
>
> Empirically, we compare the shifted MSA with various attentions, i.e., space attention (conventional self-attention, 77.02%), time attention [4] (77.62%), and concatenated feature from space and time attentions [4] (77.35%) with fixed other components in SCT-S on the Kinetics-400 dataset, which demonstrates the advantages of explicitly effective motion modeling in the shifted attention. The attention map visualization in Fig. 3 also verifies the effective motion capture of the main object in the video.
>
> * More experimental results
>
> We further conduct experiments on SSv2 and Diving-48, which are more dynamic datasets and rely heavily on the temporal dimension. Our SCT-L with Kinetics-600 pretrained model obtain 68.1% and 81.9% accuracy on the two datasets, respectively, compared with TEA [25] (65.1%, N/A), SlowFast [15] (61.7%, 77.6%), ViViT-L/16x2 [3] (65.4%, N/A), TimeSformer-L [4] (62.4%, 81.0%), and MViT-B, 32x3 [13] (67.8%, N/A). Our SCT-L achieves the best Top-1 accuracy on the two datasets. We will add VATT and update Table 3 in the supplementary.
>
> In Summary, our work has addressed several painpoints of video spatio-temporal modeling using Transformers, and we have clearly demonstrated the effectiveness of each component through extensive ablation studies. Compared with previous state-of-the-art approaches on various datasets, our method consistently achieves better accuracy with relative low FLOPs. We believe our paper can be an important work for spatio-temporal learning and video classification.
>
> ## Reference
> [3] Anurag Arnab, Mostafa Dehghani, Georg Heigold, Chen Sun, Mario Luciˇ c, and Cordelia ´ Schmid. Vivit: A video vision transformer. arXiv preprint arXiv:2103.15691, 2021. \
> [4] Gedas Bertasius, Heng Wang, and Lorenzo Torresani. Isspace-time attention all you need for video understanding? ICML, 2021. \
> [7] Joao Carreira and Andrew Zisserman. Quo vadis, actionrecognition? a new model and the kinetics dataset. In proceedings of the IEEEConference on Computer Vision and Pattern Recognition, pages 6299–6308, 2017. \
> [12] Dosovitskiy et al. An image is worth 16x16 words: Transformers for image recognition at scale. ICLR, 2021. \
> [13] Haoqi Fan, Bo Xiong, Karttikeya Mangalam, Yanghao Li, Zhicheng Yan, Jitendra Malik, and Christoph Feichtenhofer. Multiscale vision transformers. arXiv preprint arXiv:2104.11227, 2021. \
> [14] Christoph Feichtenhofer. X3d: Expanding architecturesfor efficient video recognition. In Proceedings of the IEEE/CVF Conference onComputer Vision and Pattern Recognition, pages 203–213, 2020. \
> [15] Christoph Feichtenhofer, Haoqi Fan, Jitendra Malik, and Kaiming He. Slowfast networks for video recognition. In Proceedings of the IEEE/CVF International Conference on Computer Vision, pages 6202–6211, 2019. \
> [25] Yan Li, Bin Ji, Xintian Shi, Jianguo Zhang, Bin Kang, and Limin Wang. Tea: Temporal excitation and aggregation for action recognition. In Proceedings of the IEEE/CVF Conference on Computer Vision and Pattern Recognition, pages 909–918, 2020. \
> [34] Prajit Ramachandran, Niki Parmar, Ashish Vaswani, Irwan Bello, Anselm Levskaya, and Jonathon Shlens. Stand-alone self-attention in vision models. In Advances in Neural Information Processing Systems, 2019.\
> [42] Vaswani et al. Attention is all you need. In Advances in Neural Informa469 tion Processing Systems, 2017.

---

> > ### Comment · Reviewer_yF1i · 2021-09-02
> > **Thanks for the response.**
> >
> > Thanks to the authors for their rebuttal. I appreciate the additional experiments included in the rebuttal. In my opinion, they validate each component and gives me more confidence in the proposed method. I am still not fully convinced by the explanation for how the shifted module models temporal information, but the quantitative results are quite impressive, especially on heavy temporal datasets (such as SSv2 and Diving-48 included in the rebuttal).
> >
> > I will increase my score, but I strongly encourage the authors to include all the ablation studies in the manuscript, as neither those experiments nor similar ones exist in the current version of the paper, and as other reviewers suggested, it is very important to understand the benefits of using each component of the model.

---

> > > ### Author Response · Authors · 2021-09-02
> > > **Response to Reviewer yF1i: Thank you for your suggestions!**
> > >
> > > We appreciate your suggestions! We will add all the ablation studies in the manuscript to help readers understand the benefits of using each component of the model.
> > >
> > > We are organizing code, and the code and trained models are being released soon.

---

### Official Review · Reviewer_zDkv · 2021-07-17

**Rating:** 8
**Confidence:** 5

**Summary:**

This paper proposes a new Transformer architecture for video modeling. There are following several components with their respective design motivation:
- First, each frame is split into a large number of tiny patches (e.g. 4x4 pixels) which are then linearly projected into token vectors. Then, the authors propose Image Local Chunk Attention to enforce locality in cross attention to ensure attention is only computed among neighboring patches in local neighborhoods. The authors argue that it's necessary to have small patches for good accuracy. And the Image Local Chunk Attention is therefore there to reduce computation costs brought by small patches and also enforce locality.
- With the features computed from Image Local Chunk Attention, the authors then apply a locality sensitive hashing (LSH) attention to approximate the true cross attention to increase efficiency.
- Now tokens are mixed with attention for each frame, the authors then propose Shifted Multi-Head Self-Attention that computes cross-attention across frames to model object motion and spatial variances.
- Finally, a special CLS token is extracted from each frame and passed into another clip encoder to produce a clip-level summarization, which is used for final classification tasks.
Overall, the authors proposed a set of methods to model both intra and inter frame relationships in videos using transformers and achieved SOTA results on several video classification tasks like Kinetics, UCF101, HMDB51 and Moment-in-Time. The authors also conducted thorough ablation to dissect the contribution of each components.

**Ethical Concerns:**

No ethical concern spotted.


**Limitations And Societal Impact:**

Yes

**Main Review:**

+ Given its impressive performance on images, it's now an impactful problem to study how to best utilize transformers for video modeling.
+ The paper has addressed several painpoints of video spatio-temporal modeling using transformers. First, it has proposed specific modules for intra (Image Local Chunk Attention) and inter (Shifted Multi-Head Self-Attention) frame modeling. Also, it has addressed the computation issue of transformers with the local chunk attention and locality sensitive hashing. I feel the authors have done a good job integrating these different techniques into a highly performant system with decent computation footprint.
+ Due to its high accuracy and relative low FLOPs, the proposed transformer architecture could be potentially very useful for the community, upon which more explorations can be carried out on video modeling.
+ The paper is well-written and with good illustration.
+ The authors have conducted thorough ablation to dissect their approach.

- As the authors have put a lot efforts to reduce computation of transformers. It would be nice to have the actual runtime (in seconds) for the proposed model compared to previous methods in Table 6.
- The authors designed Image Local Chunk Attention to induce locality in attention and reduce computation. It resembles the utility of convolutions. How does this compare to directly employing convolutions?
- What's the difference between "Frame Rate" and "Frame Stride" in Table 2?
- L286-7, "including the number of shifted MSA layers used in each image chunk self-attention" I'm confused about what this line means.

######### POST REBUTTAL #########

I thank the authors for their feedback. The response has clarified most of my questions, thus I will keep my rating to accept the paper.



**Time Spent Reviewing:**

6

---

> ### Author Response · Authors · 2021-08-10
> **Response to Reviewer zDkv: Thank you for your great questions and suggestions!**
>
> We greatly appreciate your questions and suggestions, and we really like them! Below are our responses.
> * We obtain the actual runtime (s) in one single NVIDIA V100 16GB GPU by averaging 50 inferences. We only evaluate models which have publicly available codes. We have TimeSformer-L [4] (0.475), X3D-L [14] (0.176), MViT-B 64×3 [13] (0.153), SCT-S (0.051), SCT-M (0.072), SCT-L (0.106).
>
> * We compare image local chunk self-attention (78.4%, 98.3%) with various convolution variants in SCT-S on the Kinetics-400 and UCF101 datasets. We have convolution (75.3%, 96.6%), convolution + bn (75.6%, 96.9%), and residual convolution block (76.8%, 97.0%), which sufficiently demonstrates the effectiveness of our image local chunk self-attention. From the perspective of receptive field size, without pooling, a four layered ConvNet with 3x3 kernel has receptive field size of 9x9, and our local chunk self-attention is able to fully model the information from 28x28 of each chunk.
>
> * The input of our model is a clip. The frame rate denotes the sampling rate of frames in one clip. Similar to TimeSformer [4], we employ multiple clips and ensemble for the inference. The frame stride is stride between two consecutive clips.
>
> * It should be “including the number of shifted MSA layers”. We will update it in the revision
>
> ## Reference
> [4] Bertasius et al. Is space-time attention all you need for video understanding? ICML, 2021. \
> [13] Haoqi Fan, Bo Xiong, Karttikeya Mangalam, Yanghao Li, Zhicheng Yan, Jitendra Malik, and Christoph Feichtenhofer. Multiscale vision transformers. arXiv preprint arXiv:2104.11227, 2021. \
> [14] Christoph Feichtenhofer. X3d: Expanding architectures for efficient video recognition. In Proceedings of the IEEE/CVF Conference on Computer Vision and Pattern Recognition, pages 203–213, 2020.

---

> > ### Comment · Reviewer_zDkv · 2021-08-25
> > **Response to author feedback**
> >
> > I thank the authors for their feedback. The response has clarified most of my questions -- It would be good if the authors could add these runtime numbers into the texts. With this, I believe it will be useful for the community to have this paper in the venue, and I will keep my rating.

---

> > > ### Author Response · Authors · 2021-09-02
> > > **Response to Reviewer zDkv: Thank you for your suggestions!**
> > >
> > > We appreciate your suggestions! We will add the runtime comparison results into the texts.

---

### Official Review · Reviewer_EmhS · 2021-07-19

**Rating:** 6
**Confidence:** 5

**Summary:**

This paper proposes a new variant of Transformer, called Shifted Chunk Transformer (SCT), for video understanding. Compared with ViT, SCT has three major contributions: 1) SCT uses a smaller patch size ($4\times 4$ instead of $16\times 16$) to extract more fine-grained features; 2) A combination of local chunk attention (i.e., self-attention within a local window) and global LSH attention is used to efficiently process the large number of tokens at each frame; 3) A frame-wise attention between adjacent frames and a clip encoder beyond the frame-level features are used for modeling temporal information. Experiments on multiple video benchmark datasets show the effectiveness and efficiency of the proposed SCT model.


Post rebuttal:
I've read the author's response and reviews from other reviewers. I think most of our questions are well addressed in the rebuttal, and I will keep my original rating "6: Marginally above the acceptance threshold".

**Limitations And Societal Impact:**

The authors adequately addressed the limitations and potential negative societal impact of their work.

**Main Review:**

The paper presents a new Transformer model that achieves the state-of-the-art results with less number of parameters and FLOPs. One of my main concerns is its novelty because most components of the model, if viewed individually, are not novel and have been explored in prior work. However, the overall technical contribution of the work is still valuable, especially the idea of using small patches for fine-grained features and the methods for modeling temporal information.

Overall, the paper is well organized and clear, but some technical details are missing:
1) In L257, what is the majority voting used for obtaining final result? Is it different from the standard averaging of multiple views?

2) In Table 4, how are the chunk size and #Tokens defined? Is the chunk size 49 comes from $7\times 7$ patches within a chunk? How about the #Tokens then?

3) How is the visualization generated in Figure 3? Where is the query position? More details should be described.

I also suggest the authors to provide more ablations and analysis:
1) One important claim of the paper is that using smaller patch size can improve the performance, and the claim is verified by ablations on ViT-B. However, the change of patch size may have different impacts for different models. I'm curious about the results of using different patch size for SCT as well, to further confirm the necessarily of using really small patch size.

2) SCT use both local and global attention at each frame. An ablation on the impact of these two types of attention is suggested to better understand the model design.

3) Why using more shift MSA block or more shifted frames will lead to worse results (Table5)? The authors should provide more discussion about the negative effect of the shifted MSA block.

4) Why using more views leads to worse results (Figure 2)? This is a bit counter-intuitive as more views provide more complete information and less noise, especially for trimmed datasets like Kinetics. More analysis on this observation is required.

**Time Spent Reviewing:**

6

---

> ### Author Response · Authors · 2021-08-10
> **Response to Reviewer EmhS (part 3/3): our novelty is unique and strong**
>
> Thank you for your question!
> Our novelty is fourfold:
> 1) To explore fine-grained intra-frame features, we construct the image local chunk self-attention to enable local and shared low-level feature extraction. Compared with previous vision related Transformers, e.g., ViT [12], the image local chunk self-attention enables an effective feature extraction from a tiny patch, which is validated to improve the spatio-temporal modeling accuracy.
> 2) To alleviate the computation and GPU memory overhead, we firstly apply the LSH attention to the vision task. The image LSH attention is able to learn image global features from all the patches in each frame. The unified structure of image local chunk self-attention and image LSH attention enables an efficient intra-frame feature extraction from local to global.
> 3) To fully consider the motion dynamics, we propose a shifted MSA block which explicitly models the motion between consecutive frames. Compared with the standard MSA, the shifted MSA improves the Top-1 accuracy by 1.5% and 1% on Kinectics-400 and UCF101 in Table 5, respectively.
> 4) The unified framework, from the image local feature extraction to the global inter-frame modeling, surpass previous state-of-the-art approaches with relative low FLOPs, including both recent Transformer based video classification and previous deep ConvNets based methods by 2.7%, 1.3%, 1.7% and 8.9% on Kinectis-400, Kinectic-600, UCF101 and HMDB51 in Table 6-7 based on RGB frames, respectively. We also conduct extensive ablation studies to validate the effectiveness of each component.
>
> ## Reference
> [12] Dosovitskiy et al. An image is worth 16x16 words: Transformers for image recognition at scale. ICLR, 2021.

---

> > ### Comment · Reviewer_EmhS · 2021-09-01
> > **Response to author comments**
> >
> > I appreciate the author's effort for answering my questions and providing more technical details. Most of my concerns are addressed in the rebuttal and I will keep my original rating. Thanks.

---

> ### Author Response · Authors · 2021-08-10
> **Response to Reviewer EmhS (part 2/3): more ablations and analysis.**
>
> Thank you for your question!
> 1) We validate different patch sizes in the SCT-S on Kinectics-400 and Diving-48. We fix other hyper-parameters and only vary the patch size. With the patch size of $2\times2$, the SCT achieves 78.4% and 80.1% Top-1 accuracy on Kinectics-400 and Diving-48, respectively. With the patch size of $8\times8$, it achieves 77.8% and 78.0% Top-1 accuracy on the two datasets. The SCT with the patch size of $2\times2$ consumes 192.53 GFLOPs which is more than twice higher than $4\times4$. The $8\times8$ takes 60.22 GFLOPs.
> 2) We validate the effect of frame-level local (image local chunk self-attention) and global attention (image LSH attention) in the SCT-S on the Kinetics-400 and UCF101 datasets. We compare the image local chunk self-attention (78.4%, 98.3%) with various convolution variants (other layers are fixed), i.e., convolution (73.9%, 94.9%), convolution + bn (74.3%, 95.0%), and residual convolution block (75.1%, 95.8%), which sufficiently demonstrates the effectiveness of our image local chunk self-attention.
> To conduct ablation studies for image LSH attention, we remove the LSH attention in the SCT-S. The SCT-S without LSH attention achieves 76.2% and 96.5% Top-1 accuracy on Kinectics-400 and UCF101. The global attention in each frame helps spatio-temporal learning.
> 3) In our experiment, the frame rate of each input clip varies from 5-10, which is 0.1-0.3s. From the perspective of the human vision system, the typical duration of persistence of vision is 0.1-0.4s. The experiment validates the best numbers of the shifted MSA and shifted frames are 1, which is consistent with our vision system and the bigger number of shifted frames could miss the motion information for some actions. From the perspective of model complexity, we have the multi-layer clip encoder after the shifted MSA to specifically model complicated inter-frame dependencies. The shifted MSA is forced to learn fine-grained motion information.
> We also empirically validate the 1 layer shifted MSA with different numbers of the shifted frames, 2, 3 and 4 in SCT-S, and obtain 78.27%, 77.43% and 77.40% on Kinectics-400, which are worse than 78.41% with only 1 shifted frame. We compare the shifted MSA with space attention [4] (77.02%), time attention [4] (77.62%), and concatenated feature from space and time attentions [4] (77.35%) with fixed other components in SCT-S, which demonstrates the advantages of the shifted attention.
> In the future work, developing multi-scale shifted MSA is an interesting topic.
> 4) The results in Figure 2 are influenced by data augmentation, e.g., random cropping, and other factors. In Fig. 2, "as the number of used views increases, the accuracy difference is reduced" because the accuracy gain is saturated. The tiny difference can be considered as fluctuation.
>
> ## Reference
> [4] Gedas Bertasius, Heng Wang, and Lorenzo Torresani. Is space-time attention all you need for video understanding? ICML, 2021

---

> ### Author Response · Authors · 2021-08-10
> **Response to Reviewer EmhS (part 1/3): more technical details.**
>
> Thank you for your question!
> 1) We obtain the final prediction by averaging the softmax probabilistic scores from these multi-view predictions. We will update it in the revision.
> 2) The chunk size is the number of local patches in each image local chunk self-attention. The chunk size of 49 comes from $7\times7$ patches within a chunk. The tiny patch size used is $4\times4$. Thus, each image local chunk self-attention takes a window of size $28\times28$ as the input without the linear pooling layer. Each frame of size $224\times224$ can be split into $64=8\times8$ chunks, which is the number of Tokens defined. The ViT [12] has no image local Transformer, and the number of Tokens is directly obtained from $(224/Patch size)^2$.
> 3) We employ the similar visualization as Figure 6 in the ViT [12], which uses Attention Rollout [52]. Specifically, we average attention weights of the shifted MSA across all heads and then recursively multiply the weight matrices of all layers, which accounts for the attentions through all layers.
>
> ## Reference
> [12] Dosovitskiy et al. An image is worth 16x16 words: Transformers for image recognition at scale. ICLR, 2021. \
> [52] Samira Abnar and Willem Zuidema. Quantifying attention flow in transformers. In ACL, 2020.

---

### Decision · Program_Chairs · 2021-09-27

**Decision:**

Accept (Poster)

**Comment:**

All reviewers recommend acceptance of this submission. The final ratings are: 6, 6, 6, 8.

All four reviewers acknowledge the strong empirical results, both in terms of accuracy as well as efficiency. The inclusion of results on motion-heavy datasets in the rebuttal period was deemed valuable and informative. It is recommended to add these results to the main paper.

Reviewers comment on the poor presentation, especially the section concerning the shifted attention. The authors should use the detailed feedback given by the reviewers in order to improve the technical discussion as well as the motivation for the approach.

The ACS agree on the recommendation of acceptance.